# Correction of Distorted Wavefront Using Dual Liquid Crystal Spatial Light Modulators

**Jiali Wu [1], Xizheng Ke [1,2,3,*], Yaqi Yang [1], Jingyuan Liang [1] and Mingyu Liu [1]**

[1] Faculty of Automation and Information Engineering, Xi'an University of Technology, Xi'an 710048, China; wujiali@stu.xaut.edu.cn (J.W.); yangyaqi@stu.xaut.edu.cn (Y.Y.); ljy@xaut.edu.cn (J.L.); liumingyu@stu.xaut.edu.cn (M.L.)

[2] Shaanxi Civil-Military Integration Key Laboratory of Intelligence Collaborative Networks, Xi'an 710126, China

[3] College of Physics and Electronics, Shaanxi University of Technology, Hanzhong 723001, China

\* Correspondence: xzke@xaut.edu.cn

**Abstract:** In space optical communication, owing to the influence of atmospheric turbulence, optical beams lose focus and become phase-distorted, which reduces the communication quality. Considering the polarization dependence of liquid crystal spatial light modulators and the dispersion effect of liquid crystal materials, the energy utilization rate of liquid crystal adaptive optics systems is low. In this study, a dual liquid crystal spatial light modulator adaptive optics system based on the GS algorithm is used to correct the wavefront distortion of a signal beam under different atmospheric turbulence intensities, and the Strehl ratio (SR) is used as the evaluation index. The simulation results show that the SR of the corrected system can be increased from 0.23, 0.41, and 0.72 to 0.77, 0.89, and 0.95, respectively. The corrected beam spot was more concentrated and the light intensity at the center of the beam spot was stronger. The experimental results show that, after the distortion wavefront is corrected by the dual liquid crystal spatial light modulator, the average gray value of the 10 × 10 pixels in the center of the spot increases from 159.3, 113.1, and 58.4 to 253.4, 247.7, and 198.3, respectively.

**Keywords:** liquid crystal spatial light modulator; wavefront sensorless detection; adaptive optics; GS algorithm

## 1. Introduction

In optical communication, owing to atmospheric turbulence, the phase of the optical beam is distorted during the transmission process, thereby reducing the communication quality. Therefore, it is necessary to correct the distorted beam at the receiving end. Currently, most distorted beam correction techniques employ adaptive optics technology [1,2].

The core device of adaptive technology is the wavefront corrector [3], and most of the literature uses a deformable mirror and a liquid crystal spatial light modulator (LC-SLM) [4]. In 2012, Roberts of the Jet Propulsion Laboratory of the California Institute of Technology, the Palomar Observatory, Cornell University, etc., used two deformable mirrors simultaneously on the PALM-300 adaptive system of the 5 m Haier Telescope to measure the low-order effects of turbulent flow. With this approach, low-order distortion and high-order distortion wavefronts are corrected. There are 241 deformable mirror drive units for low-order distortion correction and 3388 drive units for high-order distortion correction, and the Strehl ratio (SR) for K-band correction reaches a correction effect of 0.8 [5]. In 2013, the APJ reported that the 585-unit deformable mirror adaptive optics system developed by the University of Arizona was applied to a Magellan II telescope with a diameter of 6.5 m that serves the Atacama Desert. The sampling frequency was 1 kHz and there was an imaging band. In addition, the shorter the working wavelength of the telescope, the higher its diffraction-limit resolution. The application of adaptive optics in the visible-light

band has higher requirements for the drive unit density of the deformable mirror, and the manufacturing difficulty is significantly increased, therefore, it is limited in practical use [6,7]. Since the 1970s, LC-SLMs have been studied for their potential use in the field of adaptive optics owing to their high cell density [8,9]. McGlamery used two 512 × 512 phase LC-SLMs to form a wavefront control system, one of which was used to reproduce the wavefront error of the light wave, and the other was used to correct it [10]. In 2002, the Maui Island Observatory in the United States applied liquid crystal adaptive optics to a telescope with an equivalent clear aperture of 1.12 m, and it completed adaptive optics imaging in the visible-light band [11], which promoted the extensive development of liquid crystal space optics globally. In 2005, the research group of the American Association of Applied Technology improved the electronic interface of the liquid crystal spatial light corrector, and the transmission speed of the driving signal was significantly increased, which could increase the error suppression of the 3 dB bandwidth of the system to 70 Hz [12], but no imaging experiments were performed. In 2010, the Institute of Optics at the Chinese Academy of Sciences verified the feasibility of liquid crystal adaptive optics at the Xinglong Observatory [13]. LC-SLMs achieve phase modulation by changing the refractive index, and have attracted widespread attention owing to their low cost, high resolution, lack of cross-linking between liquid crystal molecules, large number of pixel units, and ability to independently program and control each pixel [14]. Huang et al. have developed a tabletop adaptive optics wavefront control system used to correct dynamic distortions. The system uses the Shack–Hartmann sensor as the wavefront sensor to measure the distortion, but the information obtained by the wavefront sensor is inaccurate due to the error of the optical path, which affects the correction effect [15]. Li et al., aiming at the problem that the liquid crystal spatial light modulator has a large number of control units and a large amount of calculation when the wavefront phase distortion compensation is directly performed, mapped the control input and the Zernike polynomial coefficients describing the wavefront distortion, which greatly reduces the dimension optimization, and effectively improves the computational efficiency [16]. Wang et al. used an interferometer for wavefront testing, described the distorted wavefront to be corrected by Zernike polynomials, and used a liquid crystal spatial light modulator to complete the static wavefront correction [17]. However, owing to the polarization dependence of LC-SLMs and the dispersion effect of liquid crystal materials, the energy utilization rate of liquid crystal adaptive optics systems is low.

Considering the low light-energy utilization rate, this study uses two LC-SLMs for wavefront correction to compensate for the energy, and to design and build an adaptive optics correction system based on the dual LC-SLM structure to achieve different corrections of signal beam distortion at atmospheric turbulence intensity. Combined with simulation analysis and experimental research, the ability of the system to correct a distorted far-field light spot is studied, which provides a theoretical basis and experimental experience for the application of dual LC-SLM structures in adaptive optics systems.

## 2. Theoretical Research

### 2.1. System Structure

An adaptive optics system without a wavefront sensor is mainly composed of a wavefront controller, wavefront corrector, and imaging detector (CCD). In application, the adaptive optics system without wavefront sensor is located at the receiving end of the free-space optical communication system, and its model schematic diagram is shown in Figure 1. After the signal light emitted by the light source is transmitted through the atmospheric channel, owing to the influence of atmospheric turbulence, wavefront distortion will occur. After the distorted beam is received at the receiving end, it will first enter the wavefront corrector. Then, after the initial correction by the wavefront corrector, when the light beam is incident on the imaging detector, the wavefront controller generates a correction control signal by performing an algorithm calculation according to the data detected by the imaging detector. It then sends it to the wavefront corrector so that the closed-loop correction of the system can be realized.

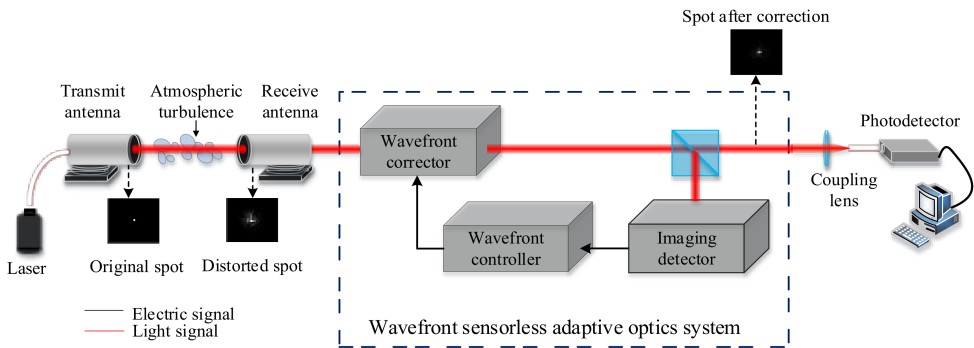

**Figure 1.** Schematic diagram of wavefront-free adaptive optics system.

## 2.2. LC-SLM Correction Principle

In this study, the reflective LC-SLM shown in Figure 2 is used as the wavefront corrector, which is mainly composed of a cover glass, transparent electrodes, nematic liquid crystals, and array electrodes [18].

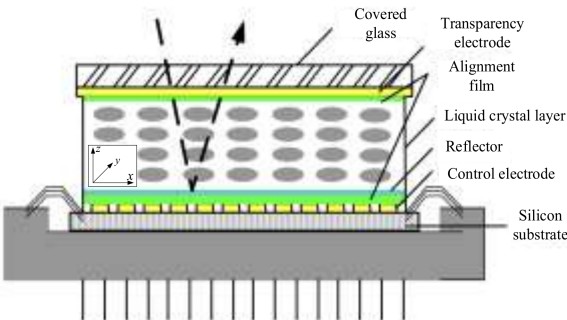

**Figure 2.** Reflective LC-SLM structure.

In the LC-SLM, let the long-axis direction of the liquid crystal molecules be parallel to the substrate $x$-axis direction and the extraordinary refractive index be $n_e$. The ordinary refractive index perpendicular to the long axis direction is $n_o$, and the linearly polarized light incident along the $z$-axis direction is under the action of an external electric field, so the liquid crystal molecules are deflected along the direction of the electric field, and the equivalent refractive index $n_{eff}$ of the liquid crystal molecules can be expressed as

$$n_{eff} = n_o n_e / \sqrt{n_o^2 \cos^2 \theta + n_e^2 \sin^2 \theta} \tag{1}$$

where $\theta$ is the angle between the long axis of the liquid crystal molecule and the $x$-axis.

After the linearly polarized light with wavelength $\lambda$ passes through the liquid crystal layer with a thickness of $d$, the resulting phase difference $\varphi$ can be expressed as

$$\varphi = \frac{2\pi}{\lambda} \int_{-d/2}^{d/2} \left[ n_{eff}(\theta) - n_o \right] dz \tag{2}$$

It can be seen that the phase difference of the outgoing light wave can be varied by changing the distribution of the electric field, thereby modulating the phase of the light wave [16,19].

The adaptive optics system corrects the beam distortion wavefront based on the principle of wavefront conjugation [20]. The working principle of the LC-SLM is to control each pixel independently by changing the addressing voltage, and the voltage of each pixel unit can be controlled by the gray value loaded onto it. The LC-SLM uses 0–255 grayscale values. When correcting the wavefront distortion, the conjugate phase grayscale image is loaded onto the LC-SLM, and the voltage is controlled by the grayscale image to realize the modulation of the phase.

### 2.3. LC-SLM Energy Compensation

Any type of polarized light can be expressed as the superposition of two linearly polarized lights with perpendicular polarization directions [21]:

$$\begin{aligned}\mathbf{E} &= E_x\mathbf{x}_0 + E_y\mathbf{y}_0 \\ &= \mathbf{x}_0 a_1 \exp[i(\alpha_1 - \omega t)] + \mathbf{y}_0 a_2 \exp[i(\alpha_2 - \omega t)]\end{aligned} \tag{3}$$

The two linearly polarized lights have a definite amplitude ratio, $a_2/a_1$, and a phase difference $\delta = \alpha_2 - \alpha_1$. In other words, the light vector of either polarization can be represented by two components along the $x$ and $y$ axes [21]:

$$E_x = a_1 \exp[i(\alpha_1 - \omega t)],\ E_y = a_2 \exp[i(\alpha_2 - \omega t)] \tag{4}$$

The amplitude ratio and phase difference of these two components determine the polarization state of the polarized light. When the common phase factor $\exp(i\omega t)$ in the above formula is omitted, the above formula can be expressed by a complex amplitude as:

$$\tilde{E}_x = a_1 \exp(i\alpha_1),\ \tilde{E}_y = a_2 \exp(i\alpha_2) \tag{5}$$

Thus, any polarized light can be represented by a one-column matrix of the two components of the light vector. This column of matrices is called the Jones vector, and is denoted as

$$\mathbf{E} = \begin{bmatrix} \tilde{E}_x \\ \tilde{E}_y \end{bmatrix} = \begin{bmatrix} a_1 \exp(i\alpha_1) \\ a_2 \exp(i\alpha_2) \end{bmatrix} \tag{6}$$

The intensity of polarized light is the sum of the intensities of its two components, i.e., [22]

$$I = \left|\tilde{E}_x\right|^2 + \left|\tilde{E}_y\right|^2 = a_1^2 + a_2^2 \tag{7}$$

Generally, the relative change in intensity is studied; therefore, the Jones vector representing polarized light can be normalized as follows:

$$\mathbf{E} = \frac{1}{\sqrt{a_1^2 + a_2^2}} \begin{bmatrix} a_1 \exp(i\alpha_1) \\ a_2 \exp(i\alpha_2) \end{bmatrix} \tag{8}$$

In addition, to enable the Jones vector to represent the amplitude ratio and phase difference of the two components, the common factor of the two components in the above formula is mentioned outside the matrix [21]:

$$\mathbf{E} = \frac{a_1 \exp(i\alpha_1)}{\sqrt{a_1^2 + a_2^2}} \begin{bmatrix} 1 \\ \frac{a_2}{a_1} \exp[i(\alpha_2 - \alpha_1)] \end{bmatrix} = \frac{a_1 \exp(i\alpha_1)}{\sqrt{a_1^2 + a_2^2}} \begin{bmatrix} 1 \\ a \exp(i\delta) \end{bmatrix} \tag{9}$$

where $a = a_2/a_1$ and $\delta = \alpha_2 - \alpha_1$. Usually, only the relative phase is considered, so the common phase factor $\exp(i\alpha_1)$ in the above formula can be omitted. Then, the Jones vector in the normalized form is obtained as

$$\mathbf{E} = \frac{a_1}{\sqrt{a_1^2 + a_2^2}} \begin{bmatrix} 1 \\ a \exp(i\delta) \end{bmatrix} \tag{10}$$

This study used the concept of orthogonal polarization. There are two columns of polarized light whose polarization states are represented by the complex amplitudes, $\tilde{\mathbf{E}}_1$ and $\tilde{\mathbf{E}}_2$. In $\tilde{\mathbf{E}}_1 \cdot \tilde{\mathbf{E}}_2{}^* = 0$, the two columns of polarized light are said to be orthogonally polarized [21], where the asterisk "*" represents a complex conjugate.

When the Jones vector is used to represent orthogonal polarization, by definition, if

$$A_1 A_2^* + B_1 B_2^* = 0 \tag{11}$$

then $\mathbf{E}_1 = \begin{bmatrix} A_1 \\ B_1 \end{bmatrix}$ and $\mathbf{E}_2 = \begin{bmatrix} A_2 \\ B_2 \end{bmatrix}$ are orthogonally polarized. Therefore, for two columns of linearly polarized light, if the vibration directions of their light vectors are perpendicular, their polarization states are orthogonal.

It can be seen that any polarization state can be decomposed into two orthogonal polarization states.

The main reasons for the low energy utilization of liquid crystal adaptive optics systems are the polarization dependence of LC-SLMs and the dispersion effect of liquid crystal materials. Because the LC-SLM achieves phase modulation by adjusting the refractive index of the light wave in the medium, the refractive index can be adjusted under an external electric field only when the polarization direction of the incident light is parallel to the alignment direction of the liquid crystal molecules [22]. The liquid crystal adaptive optics system can only work in the state of polarized light, and only after being filtered by the polarizer can it participate in the correction imaging, and the light energy will lose most of it.

Based on the above polarization theory, this study uses two LC-SLMs to correct polarized light whose polarization directions are perpendicular to each other, and it then combines the beams to complete the correction of the distorted wavefront. Thus, the energy loss caused by the modulation of the polarized light by the liquid crystal can be compensated. A schematic of the experimental setup for the wavefront correction of the dual LC-SLM structure is shown in Figure 3, wherein L1, L2, and L3 are lenses, P1, and P2 are polarizers, BS is a beam-splitter prism, and PBS is a polarized beam-splitter prism.

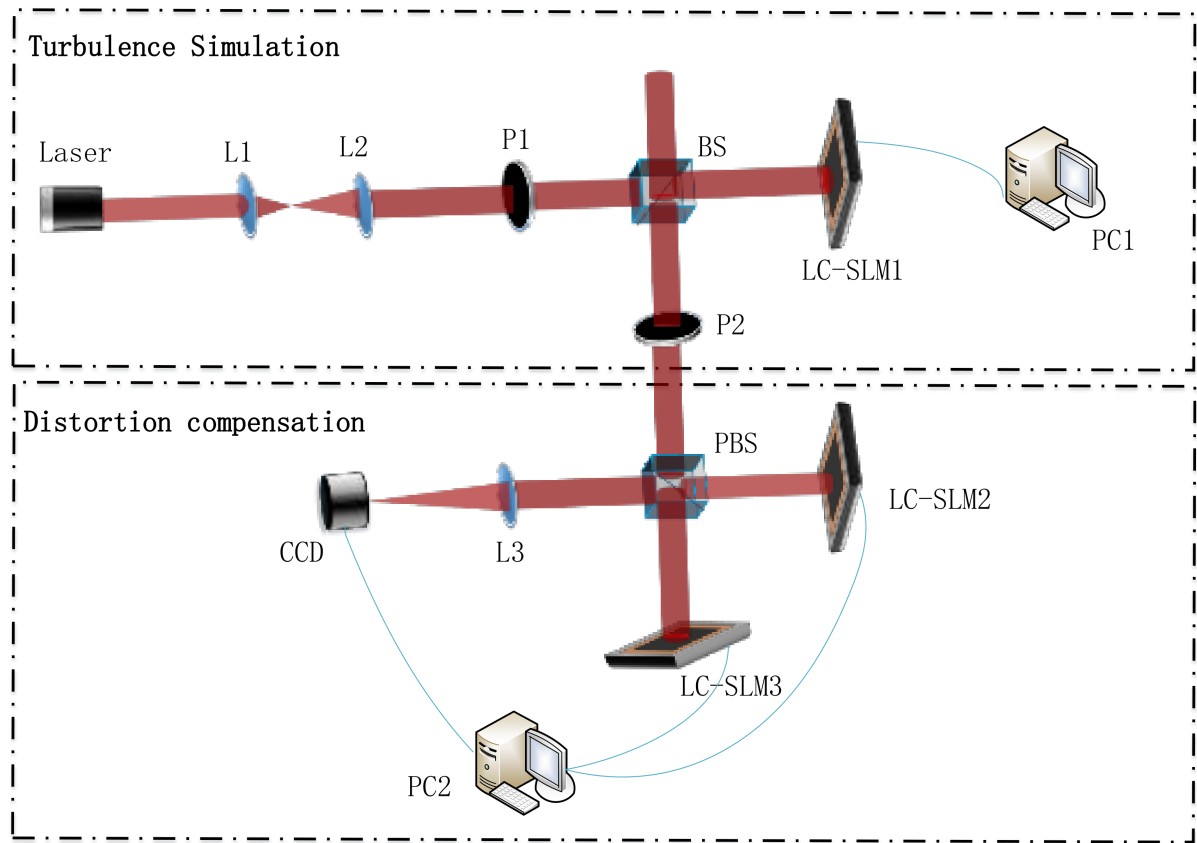

**Figure 3.** Diagram showing operating principle of dual liquid crystal spatial light modulator structure for wavefront correction.

The experimental system is divided into two parts: an atmospheric turbulence simulation and wavefront distortion compensation. (1) Atmospheric turbulence simulation: Load the simulated atmospheric turbulence phase screen onto LC-SLM1, and adjust polarizer

1 and polarizer 2 to turn LC-SLM1 into a pure phase state. Wavefront distortion occurs when the laser passes through LC-SLM1. (2) Wavefront distortion correction: The distorted parallel light is divided into two polarized lights whose polarization directions are made perpendicular to each other by a polarizing beam splitter, and they are corrected by two LC-SLMs. The two paths of light corrected by the LC-SLM are combined using a polarizing beam splitter and sent to a CCD camera for imaging. After CCD imaging, it is sent to the PC end. The PC end first reconstructs the distorted wavefront through the algorithm according to the phase distortion information of the simulation, and then it calculates the conjugate phase using the algorithm and generates a grayscale image. The PC terminal loads the conjugate phase grayscale image onto the LC-SLM, thereby correcting and compensating for the distorted phase.

### 2.4. GS Algorithm

In this paper, the GS phase-recovery algorithm is used to calculate the phase of the distorted wavefront, and the conjugate phase backpass process of the distorted wavefront is then calculated according to the phase conjugation principle to generate the corresponding grayscale image, which is loaded onto the LC-SLM to realize the transformation of the distorted wavefront.

The principle block diagram of the GS algorithm is shown in Figure 4, and its specific description is as follows [23]:

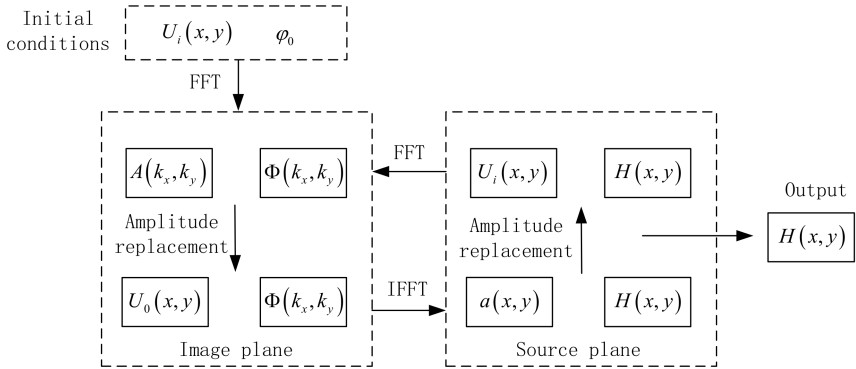

**Figure 4.** Principle block diagram of GS algorithm.

(1) Select the ideal optical field amplitude $U_i(x,y)$ without wavefront distortion as the amplitude of the input optical field, select the ideal phase $\varphi_0$ as the initial random phase, and the two forms of the optical field $U_i(x,y)\exp(i\varphi_0)$ as the input optical field for the diffraction calculation.

(2) Calculate the diffraction transmission of the light field $U_i(x,y)\exp(i\varphi_0)$ to obtain its transform-domain amplitude spectrum $A(k_x,k_y)$ and phase spectrum $\Phi(k_x,k_y)$.

(3) Replace $A(k_x,k_y)$ with the distorted beam amplitude spectrum $U_0(x,y)$ to obtain a new complex amplitude $U_0(x,y)\exp(i\Phi(k_x,k_y))$ for the light field.

(4) Perform the inverse diffraction operation on the light field $U_0(x,y)\exp(i\Phi(k_x,k_y))$ to obtain the spatial domain amplitude spectrum $a(x,y)$ and phase spectrum $H(x,y)$.

(5) Replace $a(x,y)$ with the amplitude spectrum $U_i(x,y)$ of the initial ideal light field to obtain the initial light-field expression $U_i(x,y)\exp(iH(x,y))$ for the next iteration of the loop. When certain iteration conditions are met or the defined number of cycle iterations is reached, the calculation is terminated, and the reconstructed beam distortion phase $H(x,y)$ can be obtained.

(6) The corresponding distortion phase of the simulated atmospheric turbulence is $D(x,y) = H(x,y) - \varphi_0$.

## 3. Simulation and Experiment

### 3.1. Evaluation Indicators

The SR is consistent with the performance evaluation criteria in specific applications regardless of the imaging system or laser emission system that is employed. Therefore, the SR is often used as a general performance evaluation criterion in the field of adaptive optics. The SR is defined as [24]

$$S_{SR} = \frac{I(x_0, y_0)}{I_0(x_0, y_0)} \tag{12}$$

where $I(x_0, y_0)$ is the peak intensity of the far-field spot of the distorted wavefront, and $I_0(x_0, y_0)$ is the peak intensity of the far-field spot of the ideal wavefront.

The closer the SR value is to 1, the better the actual correction result of the spot will be. In adaptive optics systems, a distorted wavefront system can be considered to be corrected well when its SR satisfies $S_{SR} > 0.8$ [25].

### 3.2. Numerical Simulation

In this study, the power spectrum inversion method was used to simulate the original distorted wavefront phase $\varphi(x, y)$ of a beam passing through atmospheric turbulence. Next, based on the phase-recovery GS algorithm, the distortion phase information was obtained from the light intensity information of the wavefront distortion. According to the principle of phase conjugation, the correction of the distorted wavefront by the LC-SLM was then simulated. The correction of the distorted light spot under different atmospheric turbulence conditions was analyzed. During the simulation, the following parameter values were set: the beam wavelength was $\lambda = 655$ nm, phase-screen width was $D = 0.4$ m, phase-screen grid number was $128 \times 128$, Gaussian beam waist radius was $w_0 = 20$ mm, transmission distance was $z = 6$ km, and phase-screen interval was 2 km. The GS algorithm was used for the phase recovery with 300 iterations. Figure 5 shows the light intensity diagrams of the spot before and after correction for strong turbulence, moderately strong turbulence, and weak turbulence.

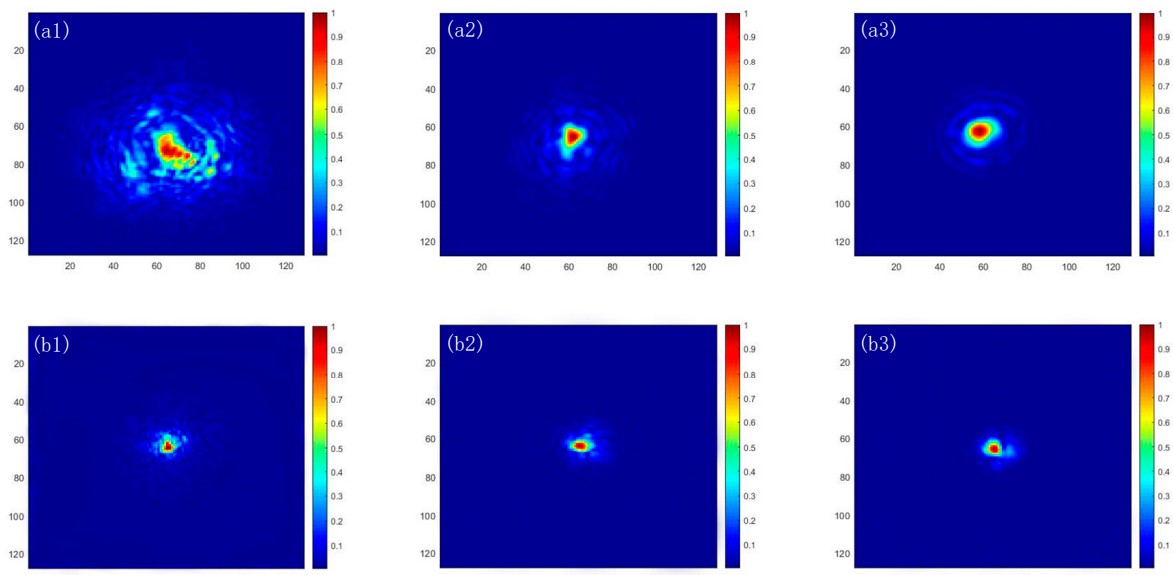

**Figure 5.** (**a1**–**a3**) Spot image before correction and (**b1**–**b3**) spot image after correction; (**a1,b1**) weak turbulence $D/r_0 = 20$; moderate turbulence in (**a2,b2**) $D/r_0 = 10$; (**a3,b3**) strong turbulence $D/r_0 = 2$.

It can be seen from the above figure that, under different turbulence intensities, the light spots before correction are fragmented and scattered, and the energy is not concentrated. With the increase in turbulence intensity, the light spots are more severely broken.

After correction by the variable LC-SLM, the light spots are more focused, and the energy becomes more concentrated.

Figure 6 shows a graph of the variation in the SR value of the system when there are 300 iterations of the algorithm under different turbulence conditions. As shown in Figure 6, in the case of strong turbulence, the SR of the system can be increased from 0.22 to 0.75; in the case of moderate and strong turbulence, the SR of the system can be increased from 0.43 to 0.89; and in the case of weak turbulence, the SR of the system can be increased from 0.73 to 0.97. The results show that the SR value can be increased after correction, and wavefront correction plays an important role.

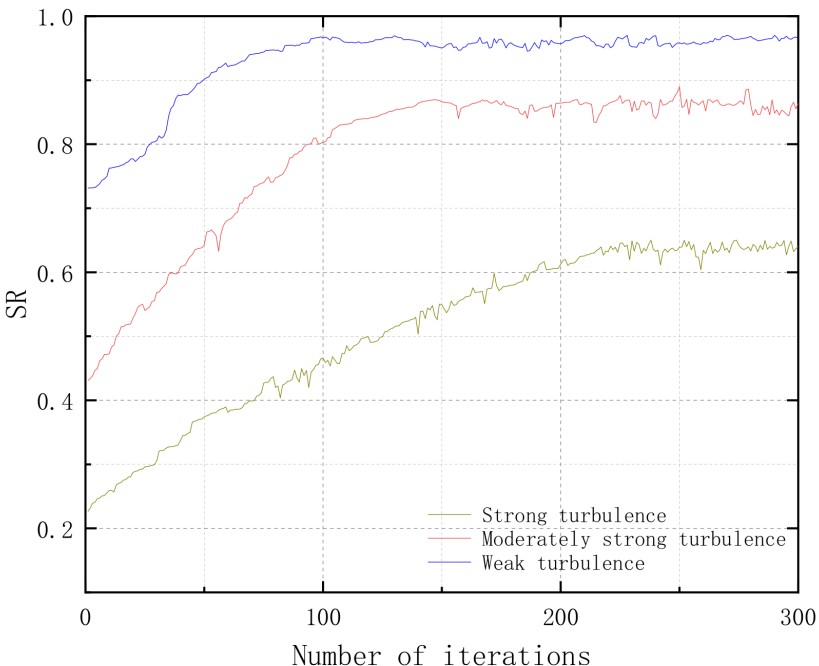

**Figure 6.** SR variation curve of the system under different turbulence conditions.

### 3.3. Experimental Study

The experimental diagram shown in Figure 7 was built according to the working principle diagram shown in Figure 3. The system mainly includes a 650 nm laser, Reallight RL-R2-SLM spatial light modulator, MV-EM series Gigabit Ethernet industrial camera (CCD camera), etc., among which L1, L2, and L3 are lenses, P1 and P2 are polarizers, BS is a beam-splitter prism, PBS is a polarized beam-splitter prism, and W1 is a 1/4 wave plate. The parameters of the equipment used in the experiment are shown in Table 1. In the experiment, the Reallight RL-R2-SLM spatial light modulator was used as the wavefront corrector. Its working wavelength was 400–700 nm, and the working wavelength of the experimental system was 650 nm, which satisfies the liquid crystal spatial light modulation, the working range of the device. The frame rate of the CCD camera is 8 fps, which is sufficient for spot acquisition. The photosensitive band of the CCD camera is 450–1100 nm, and the wavelength of the light source used in the experiment is 650 nm, which is sufficient to meet the experimental requirements.

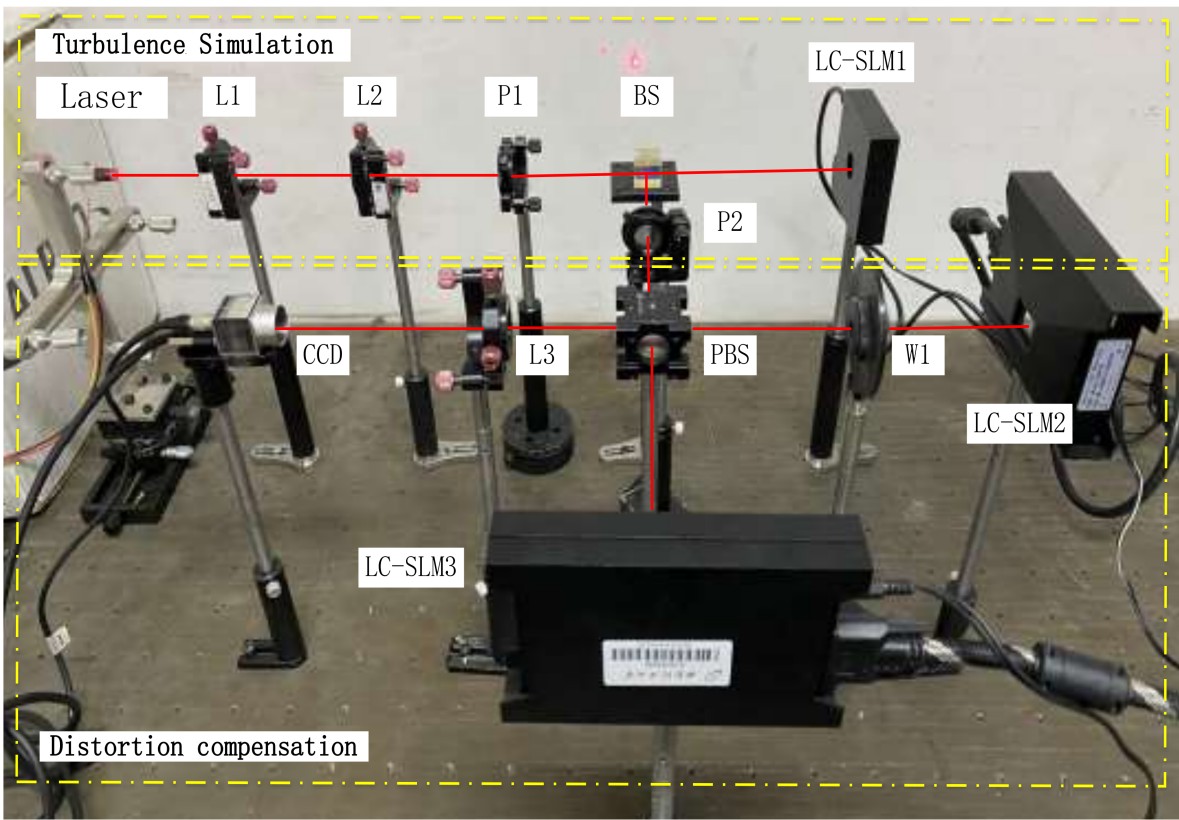

**Figure 7.** Experimental optical path diagram.

**Table 1.** Parameters of the equipment used in the experiment.

| Equipment | Model | Parameters |
|---|---|---|
| LC-SLM | RL-SLM-R2 | Target surface size: 0.78″; cell size: 12.3 μm; reflectivity: >70% ; refresh rate: 60 Hz; data interface: VGA/DVI; phase modulation capability: 0~2 $\pi$ @532 nm; working wavelength: 400~700 nm |
| CCD | MV-EM series Gigabit Ethernet Industrial Camera | Interface: GigE; pixel size: 5.5 μm × 5.5 μm; frame rate: 8 fps; exposure time: 300–1,000,000 μs; exposure method: frame exposure; synchronization mode: external trigger or continuous acquisition |

In the experiment, a CCD camera was used to collect the distorted light spot and corrected light spot images, which were then compared and analyzed. The light spot image before and after correction is shown in Figure 8.

As shown in Figure 8, after correction, the far-field-distorted light spot changes from a divergent state to a convergent state, and the intensity value at the center of the light spot is enhanced. The average gray value of the 10 × 10 pixels in the center of the spot increases from 159.3, 113.1, and 58.4 to 253.4, 247.7, and 198.3, respectively.

The brightness of the light spot was significantly improved, which was similar to the results of the simulation experimental data, and is in agreement with the theoretical analysis, and the correction effect was good.

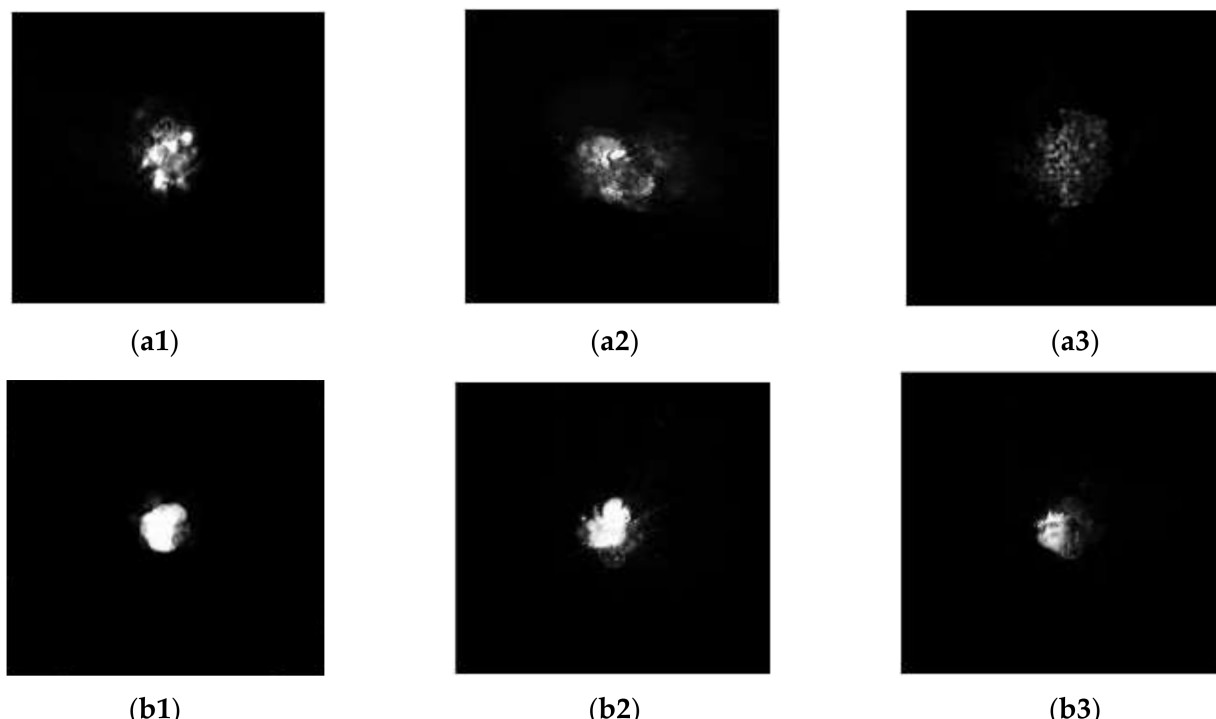

**Figure 8.** (**a1**–**a3**) Spot image before correction and (**b1**–**b3**) spot image after correction; (**a1**,**b1**) weak turbulence $D/r_0 = 2$; moderate turbulence in (**a2**,**b2**) $D/r_0 = 10$; (**a3**,**b3**) strong turbulence $D/r_0 = 20$.

## 4. Conclusions

This study focuses on the turbulent wavefront correction effect of a wavefront sensor-less adaptive optics system with a dual LC-SLM structure based on the GS algorithm. The results show that when the laser is affected by turbulence, the far-field spot exhibits speckle and centroid drift, and with an increase in the transmission distance, the influence on the laser becomes increasingly significant. The simulation results show that, after the distortion wavefront is corrected by the dual LC-SLM system based on the GS algorithm, the SR of the system can be increased from 0.23, 0.41, and 0.72 to 0.77, 0.89, and 0.95, respectively. The corrected spot becomes more convergent and the central light intensity is enhanced. The experimental results show that after the distortion wavefront is corrected by the dual LC-SLM, the average gray value of the $10 \times 10$ pixels in the center of the spot increases from 159.3, 113.1, and 58.4 to 253.4, 247.7, and 198.3, respectively. Therefore, a system based on a dual LC-SLM structure can effectively correct wavefront distortion caused by atmospheric turbulence.

**Author Contributions:** Conceptualization, X.K.; methodology, X.K., J.W. and Y.Y.; software, Y.Y.; validation, Y.Y. and J.L.; formal analysis, M.L.; investigation, J.W.; resources, J.W. and Y.Y.; data curation, J.W. and Y.Y.; writing—original draft preparation, J.W. and Y.Y.; writing—review and editing, J.W. and J.L. All authors have read and agreed to the published version of the manuscript.

**Funding:** The Key Industrial Innovation Chain Project of Shaanxi Province [grant number 2017ZDCXL-GY-06-01, 2020ZDLGY05-02]; the Xi'an Science and Technology Planning Project [grant number 2020KJRC0083]; and the Scientific Research Plan Projects of Shaanxi Education Department (18JK0341).

**Institutional Review Board Statement:** The study did not require ethical approval.

**Informed Consent Statement:** The study did not involve humans.

**Data Availability Statement:** The data that support the findings of this study are available from the corresponding author upon reasonable request.

**Conflicts of Interest:** The authors declare no conflict of interest.

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
