# Peer review of "Correction of Distorted Wavefront Using Dual Liquid Crystal Spatial Light Modulators"

_photonics, doi:10.3390/photonics9060426_

Round 1

Reviewer 1 Report

In this paper, SLM is used to simulate atmospheric turbulence and produce beam distortion. The double SLM method was used to correct the distorted wavefront, which solved the energy loss caused by the liquid crystal modulating polarized light. Compared with single SLM correction, double SLM correction method makes full use of light energy and avoids most energy loss, which has research significance for SLM phase modulation and other applications.

But there are still a few questions or changes that need to be made, as follows:

1. Does the turbulence wavefront distortion under SLM load have only a fixed value or change in real time? What are the Cn2 or R0 corresponding to strong, medium and weak turbulence in this paper?

2. How is the strel ratio of the spot before and after correction calculated in this paper, and how is the peak intensity of the ideal wavefront far field calibrated?

3. Why is the waist radius of gaussian beam set to 20mm during the simulation? Why are the width and distance of phase screen set to 0.4m and 2km? Do these match the experimental Settings?

4. In FIG. 9, after the distortion compensation part is reflected by PBS and transmitted beam splitting, how do the two orthogonal linearly polarized beams, which are reflected by SLM2 and 3 respectively, converge to CCD

Reviewer 2 Report

The paper entitled “Correction of Distorted Wavefront Using Dual Liquid Crystal Spatial Light Modulators” presented by Jiali Wua, Xizheng Ke, Yaqi Yang, Jingyuan Liang and Mingyu Liu presents a crucial problem in communications and optical signals. The presented approach is valid, I, in quality of reviewer, have a couple of comments/advice to improve their work and make it publishable on Photonics.

1) The author in the introduction can include some references about the use of nano structures or active systems able to modify or preserve features incoming beams also without or instead of the use of spatial light modulator:

i.e.  doi.org/10.3390/photonics8030065

2) I also would like to advice to merge Figure 5, 6 and 7 in a single figure with multiple panels. In this way the reader immediately can see the differences and the correction as function of the field spot at different distances.

3) Figure 9 is redundant with Figure 3, and about the way to refer to Fig. 3 is not clear what the authors would communicate saying “The structure of the double liquid crystal is shown in Figure 3. ” I advice to substitute double liquid crystal(s) with double SLM devices. The authors can merge also Figure 9 and Figure 3, and use the figure 3 as a representative flat scheme of the entire experimental setup. Also the real experimental setup of Figure 10 can be a panel of the same Figure.

4) The used equipment reported in Table 1 can be included in a proper section about the experimental procedure used to perform the experiment.

5) In the conclusions the authors remark as follow “The results show that when the laser is affected by turbulence, the far-field spot exhibits speckle and centroid drift, and with an increase in the transmission distance, the influence on the laser becomes increasingly significant.” A light that pass through air or just impinges on a SLM cannot produce spekcles, the physical condition to make a speckle are completely different. Please, see the following Nature doi.org/10.1038/37757. Anyway, you the authors can fix this in the text here and there, and explicitly say that the light beam when perturbated can changes its features and shape also from the presence of turbulences in air.  

6) I would like to advice a full check of the text.

Reviewer 3 Report

The contents of the paper look good. I will suggest the following items be addressed before publication.

1.      To allow the readers to understand the improvement achieved by the authors, please add in the introduction or conclusion section a table of comparison between this study and other papers.

2.      Figure 1 is not explicitly mentioned in the text.

3.      Figure 1: What exactly is the receive antenna? Also, what is the transmit antenna? How and with what components are they constructed, respectively?

4.      Figure 2: z axis is not indicated.

5.      Figures 5~7: What model is used to find the far-field patterns? Fraunhofer diffraction? Or, are the patterns obtained using a commercial application program?

6.      Figures 9 & 10: What do L1, P1,… etc., stand for? Please explain in the figure caption.

7.      Figure 10: Please label the turbulence simulation and distortion compensation portions as in Figure 9.

Round 2

Reviewer 2 Report

The authors answered to all my comments/ advice and the paper now is suitable for publication in Photonics Journal. I support its publication.